# Influence of Different Alkali Sulfates on the Shrinkage, Hydration, Pore Structure, Fractal Dimension and Microstructure of Low-Heat Portland Cement, Medium-Heat Portland Cement and Ordinary Portland Cement

**Yang Li [1,2], Hui Zhang [1], Minghui Huang [3], Haibo Yin [1], Ke Jiang [1], Kaitao Xiao [1] and Shengwen Tang [4,5,6,7,8,*]**

1   Changjiang River Scientific Research Institute, Changjiang Water Resources Commission, Wuhan 430010, China; liyang@mail.crsri.cn (Y.L.); zhanghui02@mail.crsri.cn (H.Z.); yinhb@mail.crsri.cn (H.Y.); jiangke@mail.crsri.cn (K.J.); xiaokt@mail.crsri.cn (K.X.)
2   School of Intelligent Construction, Wuchang University of Technology, Wuhan 430223, China
3   Baihetan Project Construction Department, China Three Gorges Construction (Group) Co., Ltd., Ningnan 615400, China; huang_minghui@ctg.com.cn
4   State Key Laboratory of Water Resources and Hydropower Engineering Science, Wuhan University, Wuhan 430072, China
5   State Key Laboratory of Building Safety and Built Environment, Beijing 100013, China
6   National Engineering Research Center of Building Technology, Beijing 100053, China
7   State Key Laboratory of Green Building Materials, China Building Materials Academy, Beijing 100024, China
8   Suzhou Institute of Wuhan University, Suzhou 215123, China
*   Correspondence: tangsw@whu.edu.cn

**Abstract:** In cement-based materials, alkalis mainly exist in the form of different alkali sulfates. In this study, the impacts of different alkali sulfates on the shrinkage, hydration, pore structure, fractal dimension and microstructure of low-heat Portland cement (LHPC), medium-heat Portland cement (MHPC) and ordinary Portland cement (OPC) are investigated. The results indicate that alkali sulfates magnify the autogenous shrinkage and drying shrinkage of cement-based materials with different mineral compositions, which are mainly related to different pore structures and hydration processes. LHPC has the lowest shrinkage. Otherwise, the effect of alkali sulfates on the autogenous shrinkage is more profound than that of drying shrinkage. Compared with the pore size distribution, the fractal dimension can better characterize the shrinkage properties of cement-based materials. It is noted that the contribution of $K_2SO_4$ (K alkali) to the promotion effect of shrinkage on cement-based materials is more significant than that of $Na_2SO_4$ (Na alkali), which cannot be ignored. The microstructure investigation of different cement-based materials by means of nuclear magnetic resonance (NMR), mercury intrusion porosimetry (MIP) and scanning electron microscope (SEM) shows that this effect may be related to the different pore structures, crystal forms and morphologies of hydration products of cement-based materials.

**Keywords:** alkali sulfates; shrinkage properties; pore structure; fractal; microstructure

## 1. Introduction

With the rapid development of human society and the economy, the service environment of modern buildings will become diversified and problematic, placing high requirements on the durability of modern cement-based materials [1,2]. Mehta [3] pointed out that the durability failure of cement-based materials was caused by internal microcracks, in which 80% were caused by non-load deformation (most of which was shrinkage), and the early deformation played a decisive role in the total deformation of cement-based materials [4–7]. Based on a large number of experimental data, Burrows et al. [8] considered that alkali sulfate content and tricalcium aluminate ($C_3A$) were the most important

and the third most important factors affecting the shrinkage and cracking performance of cement-based materials, respectively.

However, for modern cement-based materials, due to the promotion of new dry calcination technology, the reduction of high-quality cement raw materials and the service environment of ocean and saline alkali land, the alkali sulfate content of cement-based materials has become difficult to control [9]. Furthermore, it is reported that the cement industry contributes 5–7% of global $CO_2$ emissions [10,11]. The continuation of massive $CO_2$ emissions would result in a series of environmental problems such as global warming and climatic change. In recent years, low tricalcium aluminate ($C_3A$) and tricalcium silicate (C3S) cement has been widely used because of its low $CO_2$ emissions [11–14]. For instance, in China, according to its mineral composition, different kinds of Portland cement could be divided into ordinary Portland cement (OPC), medium-heat Portland cement (MHPC) and low-heat Portland cement (LHPC). Of these, the content of $C_3A$ in MHPC and LHPC is not more than 6%, and that in OPC is between 5% and 10%; furthermore, the content of C3S in MHPC is not more than 55% and that in OPC is between 50% and 60%. These differences will inevitably lead to different influences of alkali sulfates on the shrinkage properties of modern cement-based materials with different mineral compositions. However, there have been few systematic studies on this topic.

With an elliptical ring cracking apparatus, He et al. [15] observed that, with the increase in the alkali sulfate content of cement-based materials, the shrinkage rate and cracking sensitivity of cement-based materials increased. Ma et al. [16] reported that alkali sulfates could increase the drying shrinkage of cement-based materials. Yang et al. [9] found that when the alkali sulfate content increased from 0.4% to 0.8%, the plastic shrinkage of cement increased obviously. However, in the study into the effect of alkali sulfates on the shrinkage of cement-based materials, the alkali sulfates were almost expressed by the equivalent $R_2O$ ($R_2O = Na_2O + 0.658K_2O$), and the differences between $Na^+$ and $K^+$ alkali sulfates metal ions were not considered comprehensively. A few studies [17,18] have shown that, when the equivalent alkali sulfate content was certain, $K^+$ was more likely to promote the shrinkage of cement-based materials than $Na^+$, and the difference was large.

It is well known that the shrinkage mechanism of cement-based materials is closely related to their pore structure. The negative pressure caused by pore water loss is the driving force of the shrinkage of cement-based materials [6,18]. Moreover, the pore size distribution, controlled by the hydration process, has a direct impact on the shrinkage of cement-based materials [19,20]. However, when characterizing the pore structure, parameters such as porosity, pore volume or pore distribution have generally been used, which is obviously insufficient for accurately characterizing the pore structure of cement-based materials with a wide pore range and extremely irregular morphology [21–23]. In recent years, based on the fractal geometry theory, the pore structure characteristics of cement-based materials have given them significant fractal characteristics [24–27]. Fractal geometry is used to comprehensively characterize the pore structure of cement-based materials, allowing the quantification and comparison of the complex pore structure and establishing a correlation with the macro properties of cement-based materials. Kim et al. [28] reconstructed the pore structure of ground granulated blast-furnace slag( GGBFS)-blended cement pastes based on a fractal method, which could be used to analyze and predict the durability and strength of cement pastes. Jin et al. [29] established the relationship between the fractal dimension and evolution process of the freeze–thaw damage of concrete at the microscopic level. Maciej et al. [30] showed that the fractal dimension could effectively characterize the temperature crack characteristics of a low alkali cement matrix modified with microsilica. Wang et al. [31–33] indicated that, compared with the porosity and the most probable aperture, the fractal dimension could better characterize the abrasion resistance, frost resistance and permeability of concrete. As discussed above, the fractal dimension can characterize the pore structure of cement-based materials and has a good correlation with their macro properties, but few works have investigated the correlations between fractal dimensions

and shrinkage properties. Based on fractal geometry, this study establishes the relationship between fractal dimensions and shrinkage properties.

Otherwise, the effects of alkali sulfates on the hydration of different minerals are different. It has been shown that alkali sulfates could accelerate the hydration of $C_3A$ to a large extent [34–36], which was mainly due to the formation of potassium gypsum, ettringite (AFt) and U phase ($C_3A \cdot CaSO_4 \cdot 2H_2O$ system containing $Na_2SO_4$) and is related to its accelerated dissolution of Al phase ($C_3A$ and $C_4AF$) [37,38]. Some studies had shown that alkali sulfates could increase the concentration of aluminum ions in the solution but hinder the hydration of $C_2S$ [39,40]. Therefore, the influence mechanism of alkali sulfates on the shrinkage of different cement-based materials, such as OPC, MHPC and LHPC, may be related to the hydration of mineral compositions [6,8,16,41,42]. However, there were few comparative studies.

In addition, alkali sulfates could change the crystal form and morphology of hydration products, which will cause the deformation of cement-based materials [43]. Qian et al. reported that when the concentration of alkali sulfates in the solution was too high, AFt would become unstable, and monosulfide calcium sulphoaluminate (AFm) and sulfate would be generated locally [36]. Furthermore, alkali sulfates can make calcium silicate hydrate (C-S-H) to form rod-like rather than ideal needle-like hydration products and cause the size of calcium hydroxide (CH) to decrease, which was considered to be the reason for the decrease of "extensibility" of cement pastes [8,44].

Considering that the alkali sulfates in cement-based materials mainly exists in the form of alkali sulfates ($R_2SO_4$, R is Na and K), this paper mainly studies the influence of different alkali sulfates ($Na_2O$ and $K_2O$) on the drying shrinkage and autogenous shrinkage of cement-based materials of different mineral compositions by adding alkali sulfates. In addition, the influence of alkali sulfates on the hydration process, pore structure and the structure of hydration products of different cement-based materials are investigated.

## 2. Materials and Analytical Methods

### 2.1. Raw Materials

In this study, three types of cement, LHPC, MHPC and OPC with the 28-day compressive strength of 42.5 megapascals (MPa) were adopted. The chemical composition of LHPC, MHPC, and OPC was measured by X-ray fluorescence spectrometer (XRF), and its physical properties were studied according to the method described in GB/T 200-2017 (Test code for LHPC and MHPC proprieties, China) and GB/T 175-2007 (Test code for OPC proprieties, China). Chemical composition, physical properties and mineral compositions of LHPC, MHPC and OPC are presented in Table 1. It can be seen that the alkali content, $Na_2O$ content and $K_2O$ content of the three types of cement were quite different. The alkali content of LHPC, MHPC and OPC were 0.40%, 0.35% and 0.68%, in which the contents of $K_2O$ were 2.4, 2.9 and 4.9 times of $Na_2O$ by weight.

Furthermore, the content of $C_2S$ in LHPC was higher than that of $C_3S$, but the content of $C_2S$ in MHPC and OPC was lower than that of $C_3S$. The content of $C_4AF$ in LHPC was higher than 15%, which was basically the same as that of MHPC, about twice that of OPC. The biggest difference among the three types of cement was that $C_3A$ content determines the early hydration process. The content of $C_3A$ in OPC was 2.4 times that in MHPC and 4.5 times that in LHPC.

The particle size distribution of LHPC, MHPC and OPC had been measured by the laser diffraction particle size analyzer with a dry process (BT-2001, Bettersize Instruments Ltd., Liaoning, China). The particle size distribution of LHPC, MHPC and OPC are given in Figure 1. It can be found that LHPC, MHPC and OPC had similar particle size distribution, OPC had more fine particles and LHPC had more coarse particles among these three types of cement according to the results of cumulative particle size distribution.

**Table 1.** Chemical, physical properties and mineral compositions of LHPC, MHPC and OPC.

| Parameters | Raw Materials | | |
|---|---|---|---|
| | LHPC | MHPC | OPC |
| **Chemicals (wt.%)** | | | |
| CaO | 59.47 | 61.91 | 61.25 |
| $SiO_2$ | 22.59 | 21.7 | 19.84 |
| $Al_2O_3$ | 3.82 | 4.21 | 4.6 |
| $Fe_2O_3$ | 5.03 | 4.79 | 2.88 |
| $SO_3$ | 1.87 | 1.91 | 2.23 |
| $K_2O$ | 0.38 | 0.35 | 0.79 |
| $Na_2O$ | 0.16 | 0.12 | 0.16 |
| $R_2O$ [a] | 0.40 | 0.35 | 0.68 |
| Physical properties | | | |
| Specific gravity $(g/cm^3)$ | 3.23 | 3.11 | 3.11 |
| Specific surface area by BET method $(cm^2/g)$ | 336 | 325 | 356 |
| Loss on ignition (wt.%) | 0.91 | 1.28 | 3.09 |
| Fineness (% retain in 45 μm) | 8.22 | 3.27 | 5.25 |
| Mineral composition (wt.%) | | | |
| $C_3S$ [b] | 32.21 | 46.53 | 57.19 |
| $C_2S$ [b] | 40.47 | 27.11 | 13.74 |
| $C_3A$ [b] | 1.62 | 3.06 | 7.32 |
| $C_4AF$ [b] | 15.30 | 14.58 | 8.77 |

[a] Alkali sulfates content $(R_2O) = Na_2O + 0.658K_2O$. [b] From Bogue analysis.

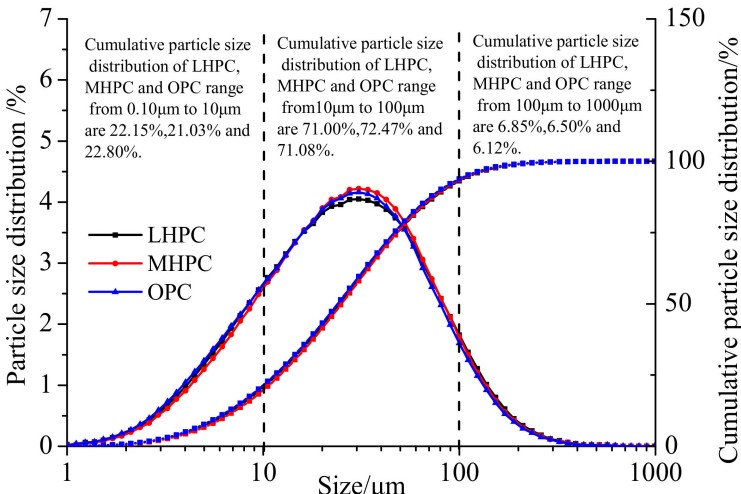

**Figure 1.** The particle size distribution of low-heat Portland cement (LHPC), medium-heat Portland cement (MHPC) and ordinary Portland cement (OPC).

The fine aggregate used in this paper was International Organization for Standardization sand. Sodium sulfate $(Na_2SO_4)$ and potassium sulfate $(K_2SO_4)$ were used to adjust the alkali content of cement.

### 2.2. Mix Proportion Design

In this work, the water-to-binder (W/B) ratio of cement paste specimens for hydration heat test, [27]Al magic angle rotation nuclear magnetic resonance ([27]Al MAS NMR) and [29]Si magic angle rotation nuclear magnetic resonance ([29]Si MAS NMR) were 0.4. Mortar specimens used for shrinkage test, mercury intrusion porosimetry (MIP) and scanning electron microscope (SEM) were prepared with a constant mass ratio (water:binder:sand = 1:2.5:5). In order to study the influence of different types of alkali sulfates on shrinkage properties and microstructure of cement-based materials containing different mineral compositions, the alkali sulfates content of cement-based materials were adjusted to 0.8% and 1.2% by

adding $Na_2SO_4$ and $K_2SO_4$, respectively. For ensuring the uniformity of additional alkali sulfates, $Na_2SO_4$ and $K2SO_4$ firstly dissolved in water and then with cement-based materials. Mix proportions and notations of mortars were summarized in Table 2.

**Table 2.** Mix proportions and notations of mortars.

| Notation | W/B | Cement Type | Cement:Sand | Added with $Na_2SO_4$ ($R_2O$), wt.% [a] | Added with $K_2SO_4$ ($R_2O$), wt.% [a] | Alkali Sulfates Content ($R_2O$), wt.% [a] |
|---|---|---|---|---|---|---|
| L0 | 0.4 | | 1:2 | - | - | 0.40 |
| L8N | 0.4 | | 1:2 | 0.92 (0.4) | - | 0.80 |
| L12N | 0.4 | LHPC | 1:2 | 1.83 (0.8) | - | 1.20 |
| L8K | 0.4 | | 1:2 | - | 1.13 (0.4) | 0.80 |
| L12K | 0.4 | | 1:2 | - | 2.25 (0.8) | 1.20 |
| M0 | 0.4 | | 1:2 | - | - | 0.35 |
| M8N | 0.4 | | 1:2 | 1.03 (0.45) | - | 0.80 |
| M12N | 0.4 | MHPC | 1:2 | 1.95 (0.85) | - | 1.20 |
| M8K | 0.4 | | 1:2 | - | 1.27 (0.45) | 0.80 |
| M12K | 0.4 | | 1:2 | - | 2.39 (0.85) | 1.20 |
| P0 | 0.4 | | 1:2 | - | - | 0.68 |
| P8N | 0.4 | | 1:2 | 0.27 (0.12) | - | 0.80 |
| P12N | 0.4 | OPC | 1:2 | 1.19 (0.52) | - | 1.20 |
| P8K | 0.4 | | 1:2 | - | 0.34 (0.12) | 0.80 |
| P12K | 0.4 | | 1:2 | - | 1.46 (0.52) | 1.20 |

[a] By weight of cement.

## 2.3. Analytical Methods

### 2.3.1. Shrinkage Behavior Measurements

Shrinkage was an important component of volume changes, possibly resulting in the occurrence of cement-based material cracks. In general, shrinkage can be classified into chemical shrinkage, autogenous shrinkage and drying shrinkage. In this study, autogenous shrinkage and drying shrinkage are mainly studied.

As shown in o Table 2, six mortars of each mix proportion were cast into 25 mm × 25 mm × 280 mm mold and cured at the environment of 90% relative humidity and $20 \pm 1\,°C$ for 24 h. After demolding, the initial lengths of specimens were measured with the help of a length comparator according to JC/T 603–2004 (Test code for mortars shrike, China). Subsequently, three specimens were moved to the chamber under temperature $20 \pm 3\,°C$ and humidity $50 \pm 4\%$ for drying shrinkage test, and the other three specimens were coated with vaseline and wrapped with preservative film to avoid evaporation of water and moved to the chamber under temperature $20 \pm 3\,°C$ for autogenous shrinkage test, the length changes of specimens were recorded at the desired ages (1, 2, 3, 4, 5, 6, 9, 11, 14, 17, 21, 25, 28, 35, 42, 49, 60, 75 and 90 days). For each kind of mix proportion, three samples were tested for each group, and the average value was reported.

### 2.3.2. Hydration Properties Measurements

The hydration heat was one of the most important hydration properties. Cement with high hydration heat was likely to generate temperature stress inside the concrete structure, which was apparently harmful to the structural safety [45–49]. Hydration heat measurements were performed using a standard calorimeter TAM AIR (TA Instruments, New Castle, DE, USA). In this study, the heat flow and accumulated hydration heat during the early stage of hydration of pastes according to the mix proportion (W/B = 0.4) were measured at $20 \pm 0.5\,°C$. Approximately 5 g of each fresh paste was placed in the glass ampoules, and the heat flow and accumulated hydration heat were recorded every 1 min for 3 days.

### 2.3.3. Pore Structure Measurements

The pore size distribution of cement-based material was measured by mercury intrusion porosimetry (MIP) using a Poremaster33 GT (Quantachrome InsDE truments, Boynton Beach, FL, USA), which consists of a low-pressure station (sensor accuracy $\pm 0.11\%$) and

a high-pressure station (sensor accuracy $\pm$ 0.05%). This instrument can measure the pore sizes ranging from 6.4 nm to 950 μm. The paste samples with a size approximately 10 mm $\times$ 10 mm $\times$ 10 mm were cut out from the center of cube specimens and prepared for MIP tests.

Prior to the testing, the samples were immersed in the anhydrous alcohol for 24 h and dried at 40 °C for 12 h in a drying oven. Considering that the shrinkage mainly occurs in the early hydration stage of cement-based material, the pastes of 3 days age were selected for MIP test.

### 2.3.4. Pore Surface Fractal Dimension

Fractal theory was an innovative method to analyze the pore structure of porous materials. Fractal dimension, which was an index derives from the fractal theory, was progressively employed to characterize the complexity and irregularity of the pore structure and macro-properties of concrete [31–33,50,51]. The pore surface fractal dimension ($D_s$) was calculated using the Zhang's fractal model based on the thermodynamic method [31–33]. The expression of the fractal model is shown in Equation (1).

$$\ln \frac{W_n}{r_n^2} = D_s \ln \frac{V_n^{1/3}}{r_n} + C \tag{1}$$

where $W_n$ is accumulated injection work; $V_n$ is the total intruded mercury content, $m^3$; $r_n$ is the radius of the smallest pore, m; $D_s$ is the pore surface fractal dimension; C is a regression constant.

The accumulated injection work can be calculated following Equation (2).

$$W_n = \sum_i^n P_i \Delta V_i \tag{2}$$

where $P_i$ is the pressure applied at the ith mercury injection, Pa; $V_i$ is the content of mercury injected at the ith mercury injection, $m^3$; n is the number of mercury injection during the MIP test.

### 2.3.5. Microstructure Measurements

[27]Al MAS NMR and [29]Si MAS NMR measurements: MAS NMR could be used to investigate the microstructure change of aluminum (Al) phases ($C_3A$ and $C_4AF$) and silicate (Si) phases ($C_2S$ and $C_3S$) in cement hydration products, reflecting the different hydration characteristics and crystal form in this study. [27]Al MAS NMR spectra were performed by Bruker Ascend 400 MHz spectrometer operating with a magnetic field of 156.3 MHz and a spinning rate of 12 kHz. A 1.0 mm zirconia rotor was used for all experiments. Samples were collected for 1024 scans, and each scan consisted of a single pulse of width 0.5 μs followed by a relaxation time of 1 s. Chemical shifts were referenced using $AlCl_3 \cdot 6H_2O$ as the external reference. [29]Si MAS NMR spectra were performed by Bruker Ascend 400 MHz spectrometer operating with a magnetic field of 119.1 MHz and a spinning rate of 6 kHz. The magnetic field intensity was 14.1 t, scanning times was 2500, relaxation time was 60 s, and 7.0 mm zirconia rotor was used for samples. Chemical shifts were referenced using tetramethylsilane as the external reference. The pastes of 3 days for MAS NMR measurements were ground and sieved through 80 μm sieve. In order to avoid carbonization, anhydrous alcohol was used as a grinding aid in the grinding process.

SEM: The morphology of hydration products of cement-based material was measured by SEM in the modes of Secondary Electron (SE) using a Quanta scanning electron microscope produced by FEI Company. The size of mortars for SEM, which were cut out from the middle of cube specimens, was approximately 2 mm $\times$ 2 mm $\times$ 5 mm. Prior to the testing, the 3-day-old samples were immersed in the anhydrous alcohol for 24 h and dried at 40 °C for 12 h in a drying oven. After that, the surfaces of samples were coated with Au to prevent charging during observation in the SEM instrument. In order to study the effect of different alkali sulfates on the morphology of hydration products of

cement-based materials with different mineral compositions, samples with 1.2% total alkali sulfates content, such as L12N, L12K, M12N, M12K, P12N and P12K, were selected. The cement without alkali sulfates was taken as the reference group.

## 3. Results and Discussion

### 3.1. The Shrinkage Properties of Cement-Based Materials with Different Types of Alkali Sulfates

The results of autogenous shrinkage and drying shrinkage of mortars with different alkali sulfates contents are given in Figures 2–4. It can be seen that alkali sulfates promote the autogenous shrinkage and drying shrinkage of LHPC, MHPC and OPC mortars, to a large extent, which cannot be ignored. In addition, the promotion effect of alkali sulfates on the shrinkage property is obvious with the increase in alkali sulfates content. This effect of the alkali sulfates in the form of $K_2SO_4$ (hereafter called K alkali) is greater than that in the form of $Na_2SO_4$ (hereafter called Na alkali). For instance, when the alkali content ranges from 0.8% to 1.2% at 28 d, the autogenous shrinkage of LHPC mortars with Na alkali increases from 38% to 41%, but that with K alkali increases from 56% to 99% compared with L0. The same conclusion can be obtained from the analysis of drying shrinkage data. However, some studies [16,52] showed that the shrinkage properties of cement-based materials had little difference when the single potassium mirabilite ($K_2SO_4$) and anhydrous mirabilite ($Na_2SO_4$) prepared by high-temperature calcination were incorporated. It can be confirmed that the effect of soluble alkali sulfates obtained on the shrinkage property of cement-based materials is more profound than that of the original alkali sulfates in cement particles, which may be attributed to the different crystal types of alkali sulfates [16,52].

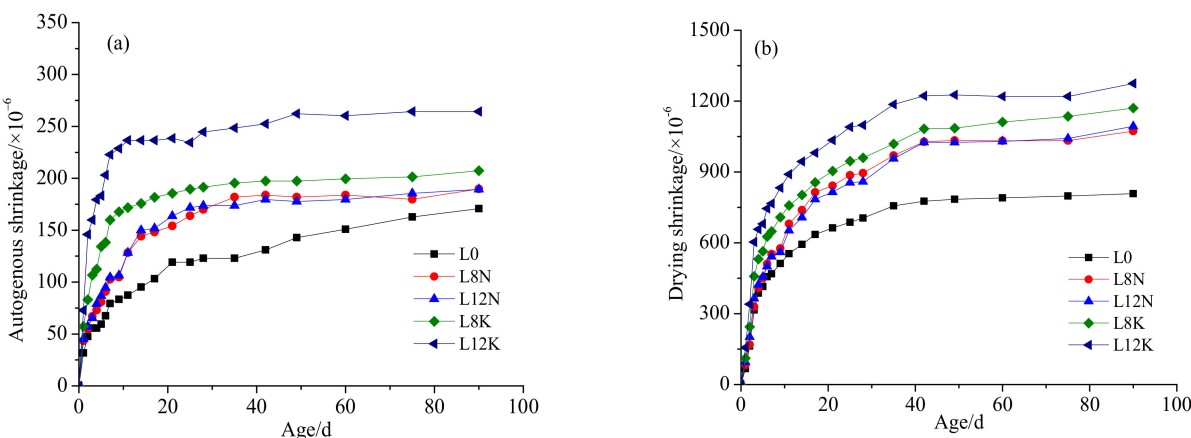

**Figure 2.** Shrinkage of LHPC mortars with different alkali sulfates contents. (**a**) Autogenous shrinkage; (**b**) drying shrinkage.

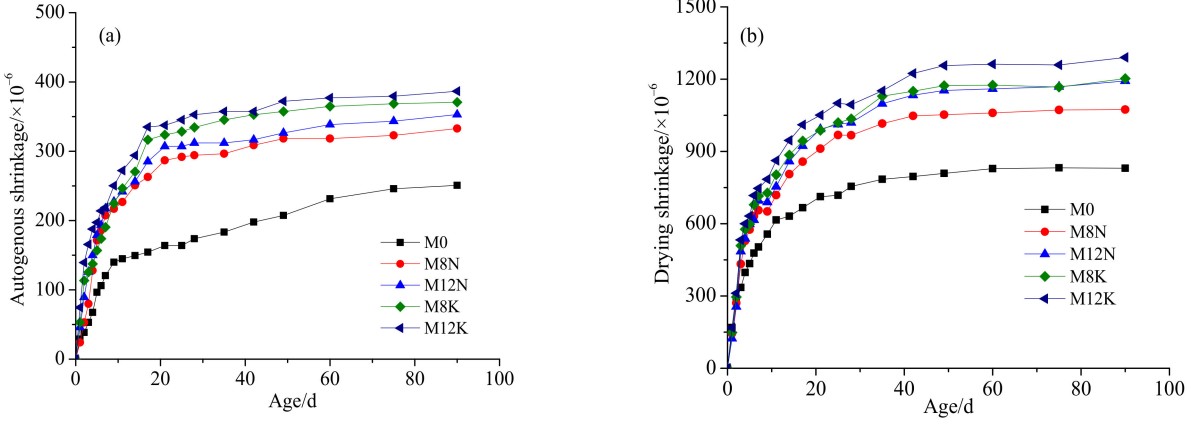

**Figure 3.** Shrinkage of MHPC mortars with different alkali sulfates contents. (**a**) Autogenous shrinkage; (**b**) drying shrinkage.

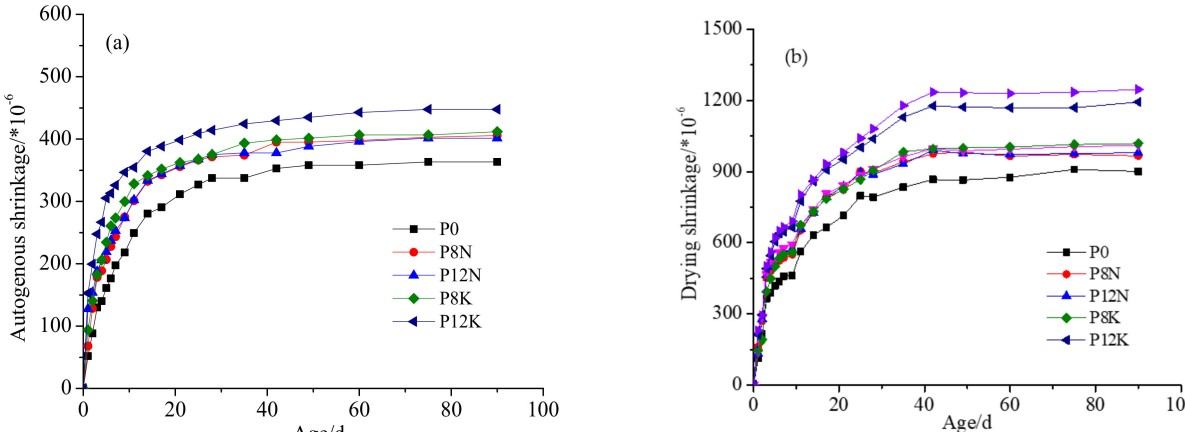

**Figure 4.** Shrinkage of OPC mortars with different alkali sulfates contents. (**a**) Autogenous shrinkage; (**b**) drying shrinkage.

Furthermore, the effect of different types of alkali sulfates on the autogenous shrinkage of cement-based materials is more profound than that on drying shrinkage. Take LHPC mortars as an example: compared with L0, when the alkali sulfate contents of mortar are from 0.8% to 1.2%, autogenous shrinkage and drying shrinkage of LHPC mortars with K alkali increase by 43% and 20% respectively at 28 d. The effect of Na alkali is consistent with K alkali. The main reason is that the driving force of autogenous shrinkage and drying shrinkage is from the internal tensile stress caused by water loss in capillary pores [53,54]. The water loss that causes autogenous shrinkage is only controlled by the hydration of cement-based materials, while the water loss for drying shrinkage includes hydration water loss and water evaporation.

It is worth noting that from Figure 2, autogenous shrinkage and drying shrinkage of LHPC mortars are small. The main reasons may lie in the different particle size distribution and mineral compositions of cement, especially the $C_3A$ phase [8]. Previous studies [55] demonstrated that alkali sulfates in the form of sulfate could greatly promote the hydration of the Al phase. By calculation, Paulini [56] showed that the chemical shrinkage values of hydration of $C_3S$, $C_2S$, $C_4AF$ and $C_3A$ were 5.32, 4.00, 11.30 and 17.85 mL/100 g. Obviously, the hydration of $C_3A$ has the largest chemical shrinkage. It is well known that most of the capillary pores, in which water loss causes the macro-shrinkage, were formed by chemical shrinkage in cement-based material [57]. In this study, though the Al phase content among LHPC, MHPC and OPC mortars are the same, the content of $C_3A$ in LHPC is 0.5 times that in MHPC and 0.2 times that in OPC mortars. This implies that few capillary pores maybe exit in LHPC mortars, resulting in small macro shrinkage. The influence of alkali sulfates on the pore structure of cement-based materials is discussed in Section 3.3.

In addition, as demonstrated in Figure 5, the ratio of autogenous shrinkage and drying shrinkage shows obvious regularity in different cement-based materials. The ratios of autogenous shrinkage and drying shrinkage of LHPC, MHPC and OPC mortars are 20%, 30% and 40% respectively. These phenomena indicate that the dry shrinkage of LHPC mortars can be reduced to a large extent by early surface curing, followed by MHPC and OPC mortars.

### 3.2. The Hydration Process of Cement-Based Materials with Different Types of Alkali Sulfates

The heat flow and accumulated hydration heat of pastes (LHPC, MHPC and OPC) with different alkali sulfates contents are shown in Figures 6–8. In general, all pastes present similar heat flow behavior up to 3 days with four distinct stages, i.e., pre-induction, induction, acceleration and deceleration stages [58–61]. Meanwhile, as noted in Figures 6a, 7a and 8a, the hydration processes of LHPC, MHPC and OPC pastes are quite different. In LHPC pastes, there seems to only a peak in the heat flow curve of L0, but in MHPC pastes and OPC pastes, the heat flow curves of M0 and P0 have obvious double peak, in which the heat flow peak of the OPC pastes are relatively sharp. These results may be attributed to

the fact that the mineral compositions of LHPC pastes, MHPC pastes and OPC pastes are different. Janse et al. [62,63] showed that the first exothermic peak was the exothermic peak of Si phase, which was mainly caused by $C_3S$ hydration, and the second one was sulfate consumption peak for $C_3A$ hydration. In LHPC pastes, the low heat of hydration because of low $C_3A$ content leads to a smooth transition from the first exothermic peak to the second one. However, in MHPC pastes and OPC pastes, the $C_3A$ content is higher than that in LHPC pastes, which makes the second exothermic peak significantly higher than the first one. Otherwise, as shown in Table 3, the maximum heat flow of L0 is 23% and 51% lower than that of M0 and P0 ones, respectively, and the 3 d accumulated hydration heat is 21% and 37% lower than that of M0 and P0, respectively, which shows that LHPC pastes have the advantage in preventing temperature cracking of mass concrete. The main reason for this may be related to the different particle size distribution and mineral compositions of LHPC, MHPC and OPC [8].

Furthermore, from Figures 5–7 and Table 3, it is found that alkali sulfates mainly promote the hydration of cement-based materials in two ways, namely, shortening the induction period and increasing the maximum hydration flow, which is consistent with the findings from the literature [17,64]. In addition, the promotion effect of K alkali is obvious compared with Na alkali. When the alkali sulfates content is 0.8%, compared with L0, accumulated hydration heat of LHPC pastes with Na alkali and K alkali are increased by 15% and 18% at 3d, respectively. This phenomenon is similar to cases of MHPC and OPC.

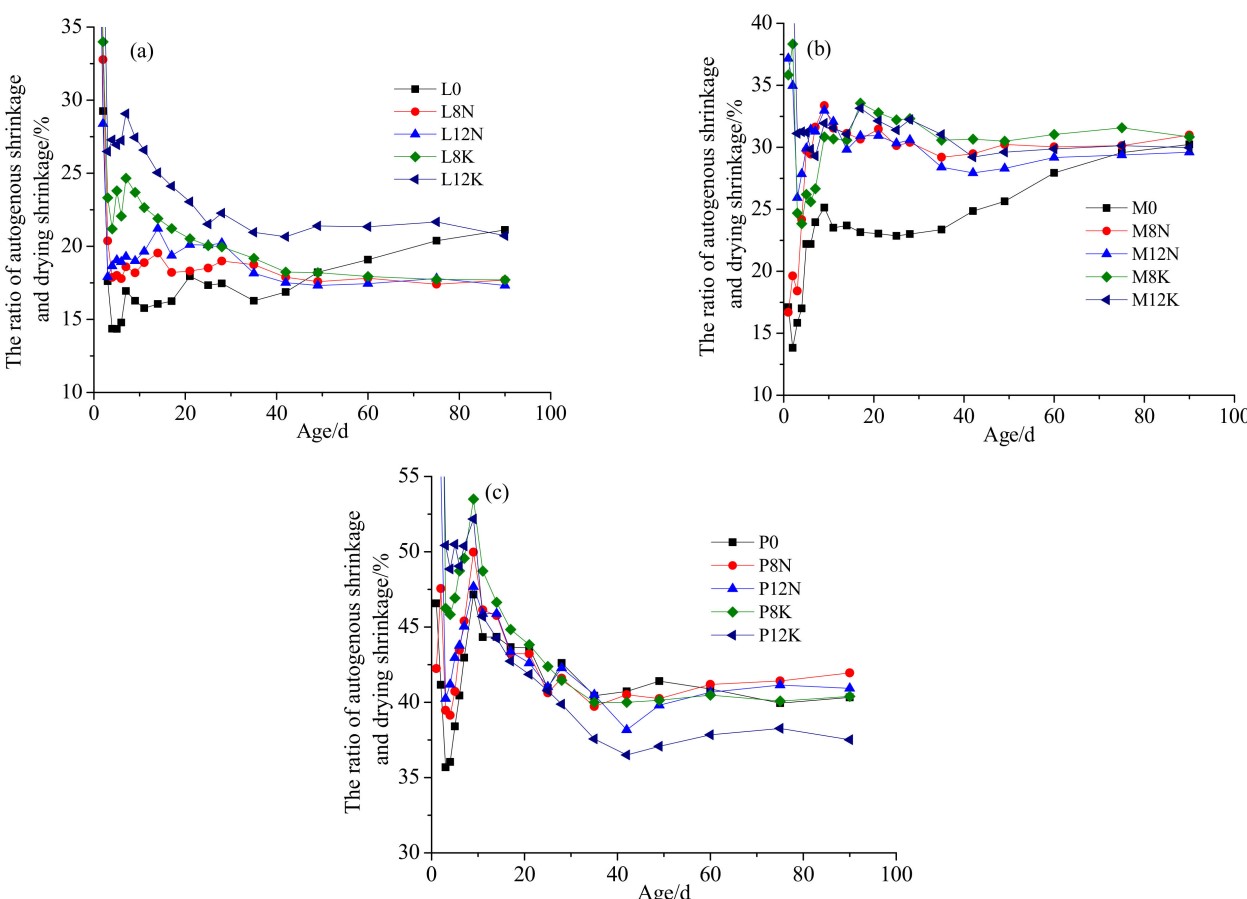

**Figure 5.** The ratio of autogenous shrinkage and drying shrinkage of mortars with different alkali sulfates contents. (**a**) LHPC; (**b**) MHPC; (**c**) OPC mortars.

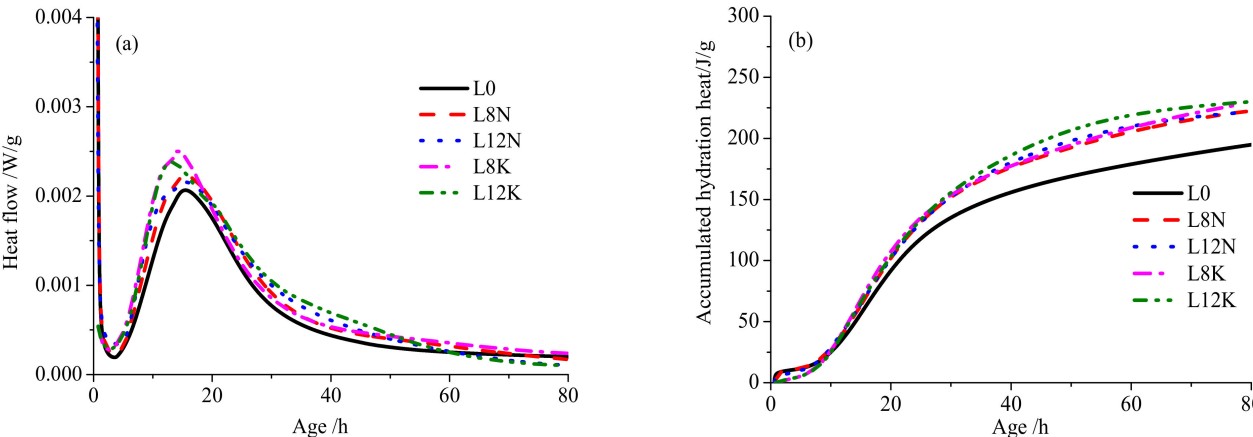

**Figure 6.** The heat flow (**a**) and accumulated hydration heat (**b**) of LHPC pastes with different alkali sulfates contents.

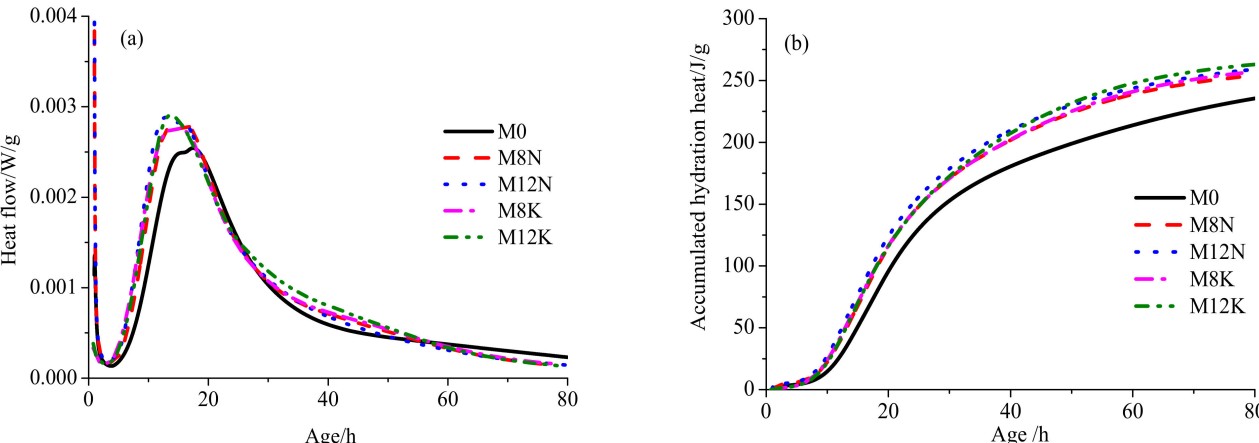

**Figure 7.** The heat flow (**a**) and accumulated hydration heat (**b**) of MHPC pastes with different alkali sulfates contents.

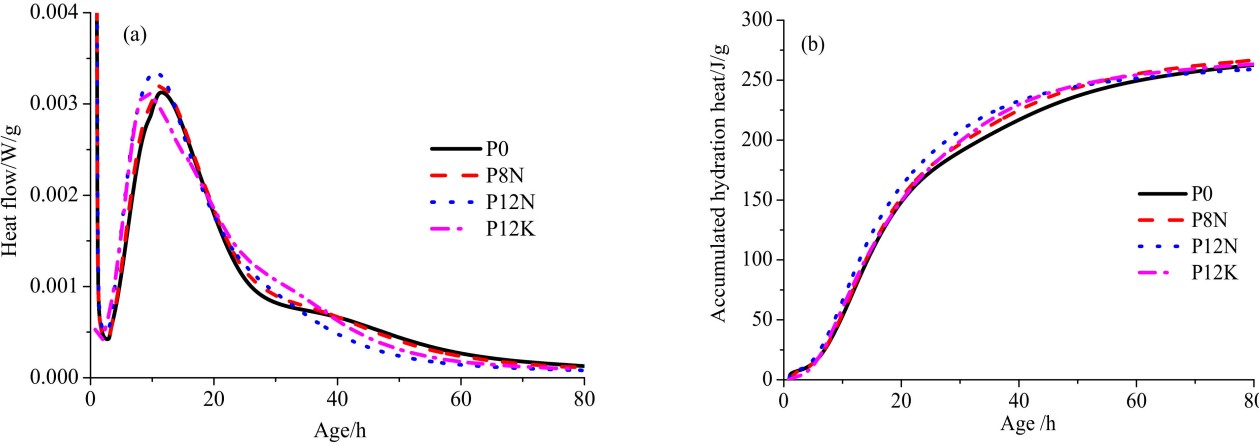

**Figure 8.** The heat flow (**a**) and accumulated hydration (**b**) heat of OPC pastes with different alkali sulfates contents.

**Table 3.** Hydration parameters of pastes with different alkali sulfates contents.

| Notation | Maximum Heat Flow/W/g | The Age of Maximum Heat Flow/h | The Age of Minimum Heat Flow/h | Accumulated Hydration Heat in 3 Days/J/g |
|---|---|---|---|---|
| L0 | 0.00207 | 15.61 | 3.58 | 188.84 |
| L8N | 0.00223 | 15.97 | 3.50 | 217.00 |
| L12N | 0.00215 | 16.25 | 4.17 | 218.29 |
| L8K | 0.00250 | 13.91 | 2.57 | 222.09 |
| L12K | 0.00238 | 12.80 | 2.74 | 226.76 |
| M0 | 0.00254 | 17.02 | 3.70 | 228.03 |
| M8N | 0.00280 | 16.00 | 3.32 | 249.57 |
| M12N | 0.00288 | 12.57 | 3.16 | 254.29 |
| M8K | 0.00279 | 14.16 | 2.43 | 252.38 |
| M12K | 0.00290 | 13.29 | 2.67 | 258.53 |
| P0 | 0.00313 | 11.37 | 2.68 | 258.35 |
| P8N | 0.00320 | 11.21 | 2.46 | 263.00 |
| P12N | 0.00334 | 10.09 | 2.36 | 256.52 |
| P12K | 0.00313 | 8.82 | 1.54 | 260.36 |

Furthermore, as shown in Table 3, the promotion effect of alkali sulfates on the hydration of LHPC pastes is greater than that of MHPC and OPC pastes. Take L8N, M8N and P8N specimens as examples, compared with the reference groups (L0, M0 and P0), accumulated hydration heat of LHPC pastes, MHPC pastes and OPC pastes increase by 28%, 21% and 5%. However, with the increase of alkali sulfates content, alkali sulfates even hinder cement hydration to some extent. For example, in L8N and L12N, the alkali sulfates content increases from 0.4% to 0.8% and then to 1.2%, and accumulated hydration heat of 3 d only increases from 28% to 29%. In P8N and P12N, accumulated hydration heat of 3 d even reduces from 5% to 2%. This is consistent with the results by Yang et al. [9]: when the alkali sulfates content increased from 0.4% to 0.8%, the cement hydration rate increased, but when the alkali sulfates content increased from 0.8% to 1.2%, the cement hydration was hindered. They believed that the hydration of cement was not simply related to the alkali sulfates content, but also to the physical and chemical properties of cement [65].

Considering the influence of cement mineral composition, the mechanism of the influence of alkali sulfates on the hydration of different cement-based materials is as follows: when alkali sulfates in the form of sulfate participate in cement hydration, it reacts with $Ca^{2+}$ in the solution to form fine $CaSO_4$ that further reacts with $C_3A$ rapidly to accelerate the dissolution of $C_3A$. At this time, the hydration is improved to a certain extent. With the increase in alkali sulfates content, when a large amount of $SO_4^{2+}$ exists, $SO_4^{2+}$ would be adsorbed on the surface of $C_3A$ due to electrostatic action [66,67]. AFt generated by rapid hydration will be also covered on the surface of non-hydrated cement particles, hindering the hydration of cement. In addition, a large number of studies [35,39,40,68] show that Al ions in solution hinder the dissolution of Si phase in cement, reducing the hydration degree of Si phase in cement. $Al^{3+}$ can adsorb on the surface of hydroxylated $C_3S$ because of a strong ion reaction of $Ca^{2+}$ and $Al^{3+}$.

*3.3. The Pore Structure of Cement-Based Materials with Different Types of Alkali Sulfates*

Figures 9–11 illustrate the curves of cumulative pore volume and pore size distribution of mortars with different alkali sulfates contents at 3 d. The pore distribution of mortars with different alkali sulfates contents at 3 d is calculated and listed in Table 4. It can be clearly seen that the most probable pore size of L0-3d is 2.2 and 2.4 times larger than that of M0-3d and P0-3d. With the increase in alkali sulfates content, the most probable pore size of LHPC, MHPC and OPC mortars decreases, which shows that alkali sulfates can refine the pore structure of cement-based materials. However, the refining effect of the same type of alkali sulfates on LHPC mortars is greater than that of MHPC mortars and OPC mortars. For instance, when the Na alkali content of cement-based materials is 1.2%, the most probable pore sizes of LHPC, MHPC and OPC mortars are reduced by 58%,

24% and 17%, respectively. In addition, from Table 4, it could be observed the refining effect of K alkali on cement-based materials is greater than that of Na alkali. In the case of LHPC mortars, the most probable pore sizes of LHPC mortars with K alkali and Na alkali are reduced by 63.0% and 58.8%, respectively. Thomas et al. [69] reported that the pore refinement of cement-based material was caused by the filling of hydration products or the collapse of large pores.

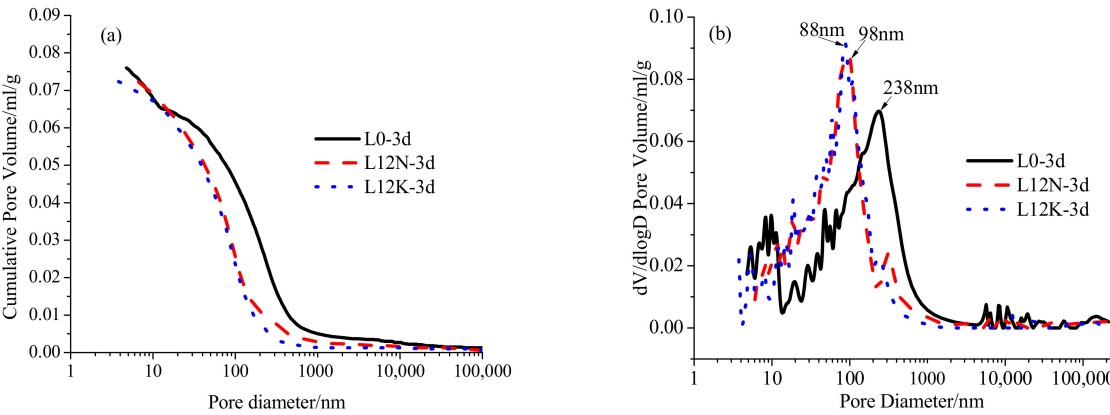

**Figure 9.** The curves of cumulative pore volume (**a**) and pore size distribution (**b**) of LHPC mortars with different alkali sulfates contents at 3 d.

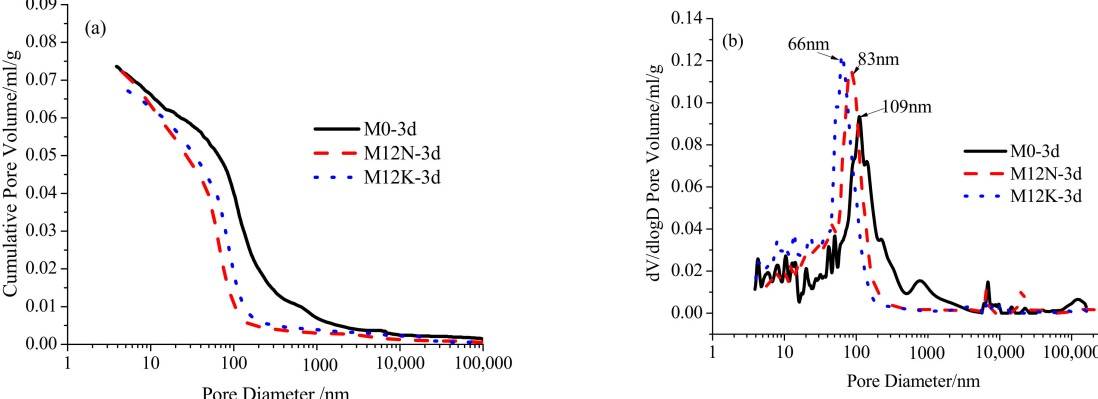

**Figure 10.** The curves of cumulative pore volume (**a**) and pore size distribution (**b**) of MHPC mortars with different alkali sulfates contents at 3 d.

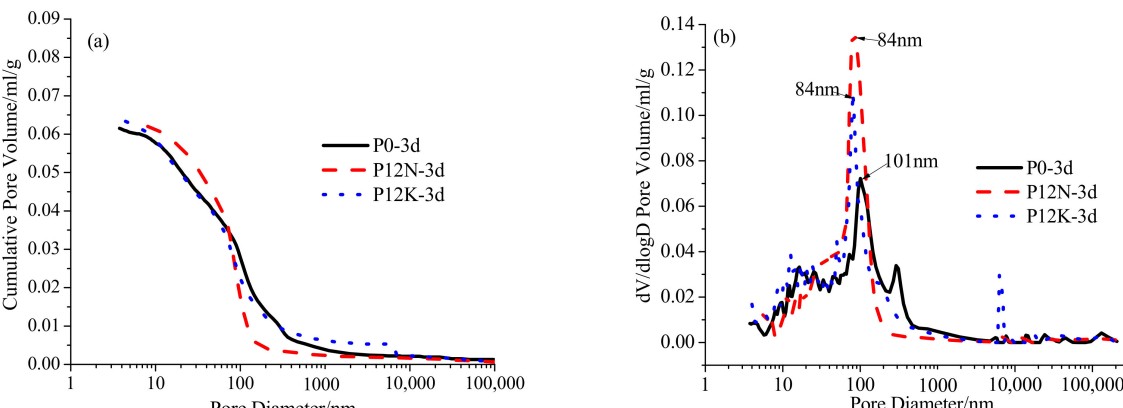

**Figure 11.** The curves of cumulative pore volume (**a**) and pore size distribution (**b**) of OPC mortars with different alkali sulfates contents at 3 d.

**Table 4.** The pore distribution and $D_s$ of mortars with different alkali sulfates contents at 3d.

| Notation | The Most Probable Aperture/nm | Porosity/% | Pore Size Distribution/% | | | | | $D_s$ |
| --- | --- | --- | --- | --- | --- | --- | --- | --- |
| | | | 5 nm~20 nm | 20 nm~50 nm | 50 nm~100 nm | 100 nm~200 nm | >200 nm | |
| L0-3d | 238 | 23.59 | 16.15 | 11.42 | 14.80 | 21.54 | 36.09 | 2.3426 |
| L12N-3d | 98 | 23.02 | 15.51 | 19.26 | 30.26 | 19.25 | 15.72 | 2.3547 |
| L12K-3d | 88 | 22.70 | 15.54 | 22.6 | 29.66 | 22.60 | 9.60 | 2.4585 |
| M0-3d | 109 | 21.99 | 14.92 | 10.6 | 18.83 | 27.89 | 27.76 | 2.2841 |
| M12N-3d | 83 | 20.76 | 15.34 | 17.57 | 34.87 | 18.57 | 13.65 | 2.3360 |
| M12K-3d | 66 | 20.52 | 23.57 | 20.50 | 40.03 | 8.79 | 7.11 | 2.3909 |
| P0-3d | 101 | 18.89 | 17.19 | 17.02 | 20.00 | 22.32 | 23.47 | 2.2793 |
| P12N-3d | 84 | 17.80 | 16.31 | 19.78 | 36.65 | 20.72 | 6.54 | 2.3017 |
| P12K-3d | 83 | 17.70 | 21.30 | 17.17 | 24.17 | 18.28 | 19.08 | 2.3170 |

In cement-based materials, the driving force of non-load shrinkage comes from the internal capillary negative pressure ($\Delta p$) caused by the evaporation of pore water. Changes of capillary negative pressure can be calculated from Kelvin equation (Equation (3)) and Laplace equation (Equation (4)) [8].

$$2\sigma\cos\theta/r = -RT\ln(RH)/Vw \tag{3}$$

$$\Delta p = 2\sigma\cos\theta/r \tag{4}$$

where $\sigma$ is the tension of the pore-fluid, mN/m; $\theta$ is the liquid-solid contact angle, °; $r$ is the radius of the interfacial meniscus, m; R is the ideal gas constant, J/(mol·K); T is the thermodynamic temperature of the system, K; Vw is the molar content of the pore fluid, $m^3$/mol; RH is the relative humidity, %. It can be seen that $\Delta p$ increases with the decrease in the internal humidity and pore diameter. In cement-based materials, an increase in cement hydration means an increase in internal water consumption, which leads to a decrease in internal humidity. Thus, as shown in Tables 3 and 4, there are two possible reasons to explain the promotion effect of alkali sulfates on the shrinkage of cement-based materials. One possible reason is the hydration promotion effect that can reduce internal humidity of cement-based materials. The other possible reason is the pore refinement effect that can reduce the pore sizes of cement-based materials.

However, the small autogenous shrinkage and large drying shrinkage of different cement-based materials are mainly caused by the internal humidity of the mortars. It was found that when the humidity was less than 85%, the cement would no longer be hydrated [70]. These phenomena indicate that the autogenous shrinkage is affected by the lower limit of 85% of the internal humidity, while the relative humidity of the drying shrinkage in this work is $50 \pm 4\%$, leading to a large driving force of drying shrinkage and large drying shrinkage.

Moreover, the mechanism of K alkali promotes the shrinkage of cement-based materials more proud than Na alkali can be revealed by the solution surface tension and pore size distribution. For the solution surface tension, previous studies [71] have shown that the alkali sulfates could effectively improve the solution surface tension, leading to a high capillary negative pressure. Taylor et al. [72,73] pointed out that the adsorption and solidification effect of C-S-H on $Na^+$ was greater than that of $K^+$, leading to the high concentration of Na alkali in the solution than that of K alkali. Therefore, the surface tension and capillary negative pressure increase in the pore solution of cement-based material contain K alkali. Furthermore, K alkali can increase the proportion of pores whose diameters are smaller than 200 nm, especially smaller than 50 nm. Taking the LHPC mortars as an example, such proportions smaller than 50 nm in LHPC with Na alkali and K alkali are 36% and 38%, respectively, which are also significant in MHPC mortars and OPC mortars. Metha [74] pointed out that in cement-based materials, pores larger than 50 nm had a great impact on strength and permeability, while pores smaller than 50 nm were on shrinkage and creep. Wang et al. [12,75,76] have got the same results. According to Equation (4), the smaller

the pore size, the larger the negative pressure of capillary, which will lead to larger macro shrinkage of cement-based materials.

### 3.4. Fractal Analysis of Pore Structure and Shrinkage Behavior of Cement-Based Materials with Different Types of Alkali Sulfates

3.4.1. Relationship between the Pore Structure and $D_s$

According to experimental data of MIP results and Equation (1), $D_s$ values of mortars with different alkali sulfates contents are calculated and listed in Table 4. It can be seen that all the values are in the range of 2.0 to 2.5. Based on the principle of the fractal theory, the $D_s$ values of pore structure are meaningful between 2.0 and 3.0 [27,77–79], indicating that the pore structure of cement-based materials with different mineral compositions has typical fractal characteristics.

As shown in Table 4, the $D_s$ values of cement-based materials increase with the increment of alkali content. At the same alkali content, the $D_s$ value of cement-based materials with K alkali is higher than that of Na alkali. $D_s$ is an index to characterize the complexity and heterogeneity of pore structure. When the $D_s$ of porous materials was equal to 2, the materials measured were a smooth plane, while the pore structure became rough and complex with the $D_s$ value approaching 3 [80,81]. Thus, combined with the shrinkage properties of cement-based materials in Section 3.1, it confirmed that alkali sulfate can make the pore structure of cement-based materials rough and complex, causing great macro shrinkage, in which the promotion effect of K alkali is more obvious than that of Na alkali. Furthermore, it should be noted that the $D_s$ values of LHPC mortars are larger than those of MHPC mortars and OPC mortars. This may be related to the hydration degree of different minerals and the distribution of different cement particles, which needs to be further studied.

In addition, the relationships between the pore structure and $D_s$ of mortars with different alkali sulfates contents are given in Figure 12. Figure 12a confirms that there is a linear relationship between $D_s$ and porosity of cement-based materials. These results are in good agreement with previous findings [31–33]. It can be also seen that in LHPC, MHPC and OPC mortars, the slope values of curves between porosity and $D_s$ are 0.0698, 0.0628 and 0.0263. This shows that the influence of alkali sulfates on $D_s$ of LHPC and MHPC mortars is greater than that on porosity. Furthermore, it can be observed from Figure 12b that there are linear relationships between $D_s$ and the pore distribution of 5–50 nm, and the wider the pore distribution of 5–50 nm, the larger the $D_s$ value. Therefore, in LHPC and MHPC mortars, $D_s$ can effectively characterize the influence of alkali sulfates on pore structure of cement-based materials.

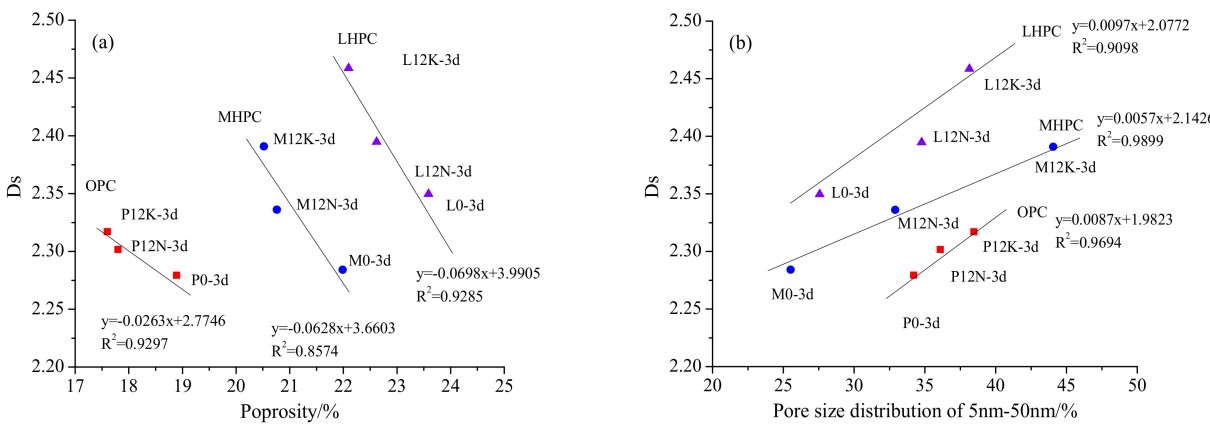

**Figure 12.** Relationship between pore structure and $D_s$ of mortars with different alkali sulfates contents. (**a**) Porosity; (**b**) pore size distribution of 5–50 nm.

### 3.4.2. Relationship between Shrinkage Behavior and $D_s$

The relationship between shrinkage properties and Ds of cement-based materials is rarely studied. In this study, the relationships between the shrinkage behavior (including the autogenous shrinkage and drying shrinkage) and $D_s$ are revealed and plotted in Figure 13. It can be seen that in the cement-based materials with the same mineral composition, the correlation $R^2$ between autogenous shrinkage and $D_s$ is larger than 0.8891, and that between drying shrinkage and $D_s$ is larger than 0.9096, demonstrating that $D_s$ can reflect the shrinkage properties of cement-based materials: the larger $D_s$ is, the larger the shrinkage is.

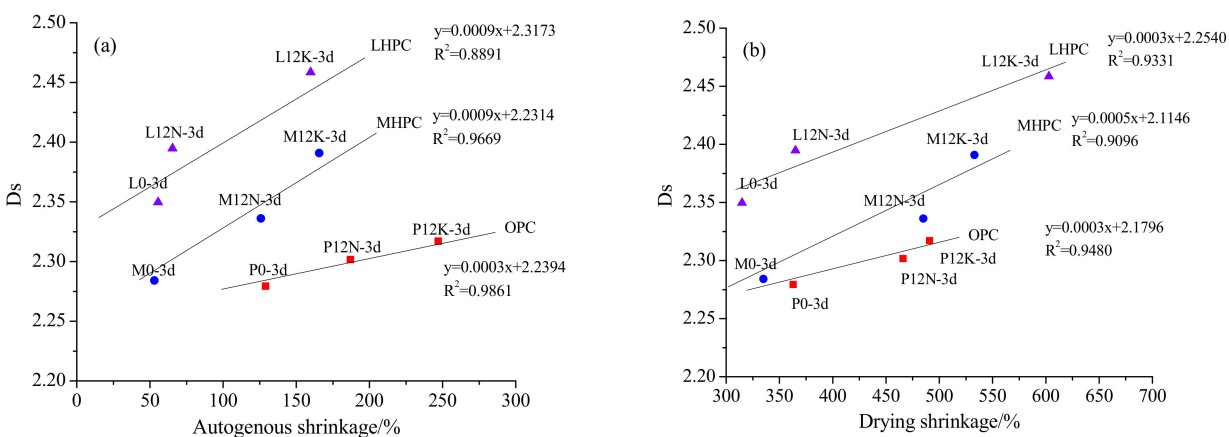

**Figure 13.** Relationship between shrinkage behavior and $D_s$ of mortars with different alkali sulfates contents. (**a**) Autogenous shrinkage; (**b**) drying shrinkage.

Furthermore, a large number of studies show that pores whose diameters are smaller than 50 nm were the main factor affecting the shrinkage properties of cement-based materials [74–76]. Figure 14 illustrates the relationship between shrinkage behavior and pore size distribution of 5–50 nm of mortars with different alkali sulfates contents. Compared with Figure 12, it can be observed that in the cement-based materials with the same mineral composition, shrinkage properties are correlated with the $D_s$ with a high $R^2$ value. For instance, in LHPC mortars, $R^2$ between autogenous shrinkage and $D_s$ is 0.8891, but that between autogenous shrinkage and pore size distribution of 5–50 nm is 0.6391. This indicates that $D_s$, as a comprehensive characterization parameter of pore structure of cement-based materials can better characterize the shrinkage characteristics of cement-based materials than the pore size distribution of 5–50 nm.

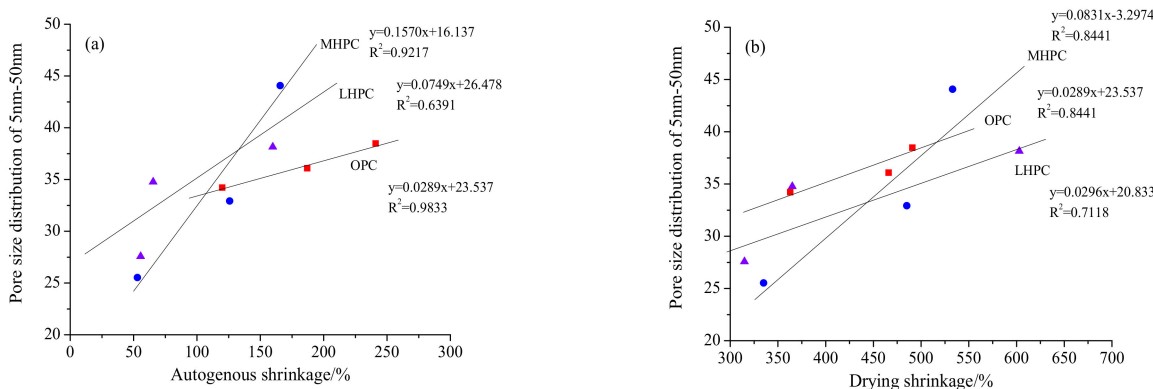

**Figure 14.** Relationship between shrinkage behavior and pore size distribution of 5 nm–50 nm of mortars with different alkali sulfates contents. (**a**) Autogenous shrinkage; (**b**) drying shrinkage.

### 3.5. The Hydration Products of Al Phase of Cement-Based Materials with Different Types of Alkali Sulfates

Considering that different types of alkali sulfates have similar effects on shrinkage, hydration process and pore structure of different types of cement-based materials, this section takes LHPC pastes as an example to explore the influence of different types of alkali sulfates on the hydration of the Al phase. The $^{27}$Al MAS NMR spectra of LHPC pastes at 3 d are shown in Figure 15. It can be seen that different contents and different types of alkali sulfates do not affect the relative chemical position of the Al phase in LHPC pastes much. The assignments of $^{27}$Al NMR components are based on previous literature [82–84]. The main peaks of chemical shift at about 86 ppm and 81 ppm are in the unhydrated Si phase (Al ($C_3S/C_2S$)) and $C_3A$. With the hydration of cement pastes, Al atom gradually migrates to the hydration products and exists in the form of four-coordination (Al (4)), five-coordination (Al (5)) and six-coordination (Al (6)). Al (4) is mainly located in the C-S-H chain and the chemical shift peak is around 75 ppm. Al (5) is mainly between the C-S-H layers, and the chemical shift peak is near 20–40 ppm. Al (6) is in the AFt, AFm and the third phase (TAH) with the chemical shift peaks of 13.1 ppm, 9.8 ppm and 5 ppm, respectively.

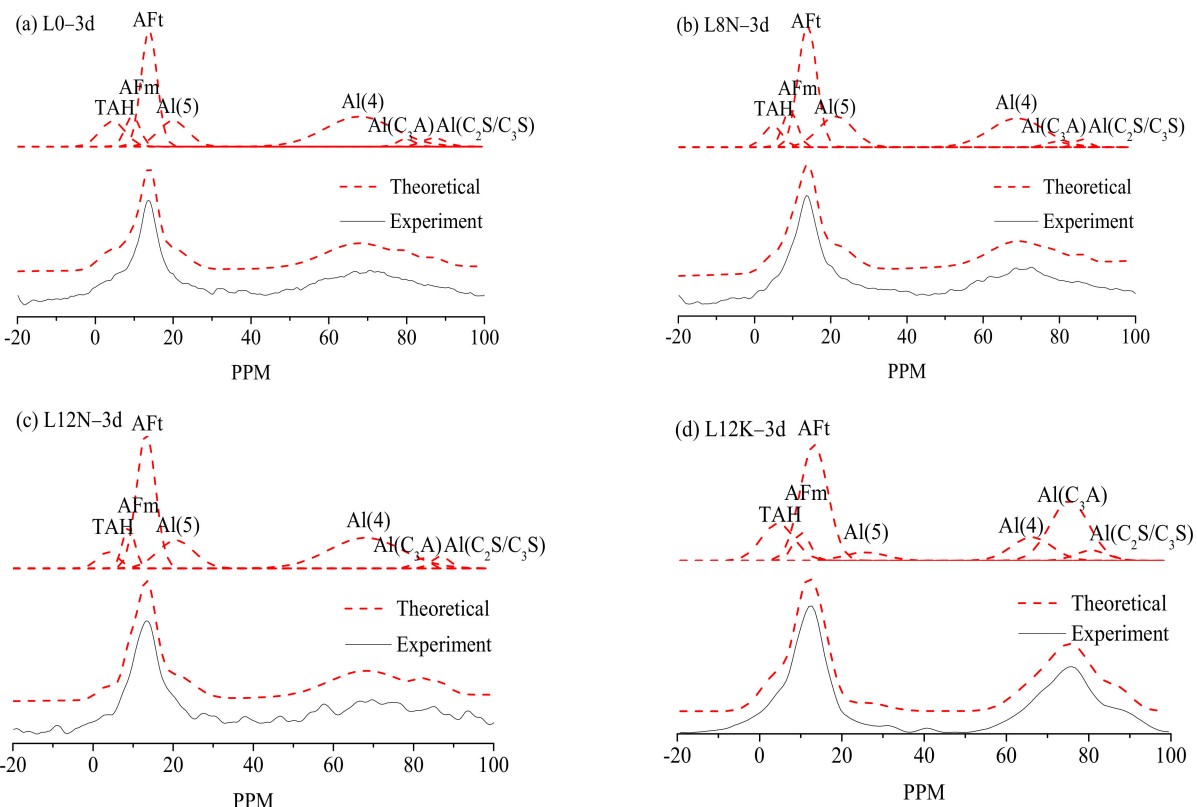

**Figure 15.** $^{27}$Al MAS NMR spectra of LHPC pastes with different alkali sulfates contents at 3 d. (**a**) LHPC pastes with alkali content of 0.4% at 3d; (**b**) LHPC pastes with alkali content of 0.8% (added with $Na_2SO_4$) at 3 d; (**c**) LHPC pastes with alkali content of 1.2% (added with $Na_2SO_4$) at 3 d; (**d**) LHPC pastes with alkali content of 1.2% (added with $K_2SO_4$) at 3d.

According to the deconvolution calculation of $^{27}$Al MAS NMR curve, the Al phase composition of hydration products of LHPC pastes with different alkali sulfates at 3 d is shown in Table 5. From Table 5, when the alkali sulfates content is smaller than 0.8%, alkali sulfates can promote the hydration of Al phase in LHPC to some extent, but with the increase in alkali sulfates content, the degree of hydration of the Al phase in LHPC decreases. For example, when alkali sulfates content increases from 0.8% to 1.2%, the content of $C_3A$ in LHPC with Na alkali decreases from 98.16% to 97.67%. K alkali shows

the same trend, but the hindrance to the hydration of $C_3A$ is obvious, such as when the alkali sulfates content is 1.2%, the hydration degree of $C_3A$ is 96.55%, even lower than 97.08% of L0. However, the decrease in the hydration degree of the Al phase in LHPC does not mean the decrease in the overall hydration degree of LHPC pastes. From Table 3, it can be seen that the overall hydration degree of LHPC with alkali sulfates is improved because the decrease in the hydration degree of Al phase in LHPC pastes accelerates the dissolution of the Si phase ($C_2S$ and $C_3S$) [35,39,40,68].

**Table 5.** Al phase composition of hydration products of LHPC pastes with different alkali sulfates at 3d.

| Notation | Al Phase Composition/% | | | | | | |
|---|---|---|---|---|---|---|---|
| | I[Al ($C_2S$ and $C_3S$)] | Al ($C_3A$)] | I[Al (4)] | I[Al (5)] | I[Al (6)] | | |
| | | | | | I(AFt) | I(AFm) | I(TAH) |
| L0-3d | 3.61 | 2.92 | 31.16 | 12.80 | 31.22 | 7.60 | 10.70 |
| L8N-3d | 1.74 | 1.84 | 32.28 | 13.48 | 35.85 | 8.47 | 6.34 |
| L12N-3d | 3.11 | 2.33 | 31.48 | 13.38 | 37.24 | 7.55 | 4.92 |
| L12K-3d | 5.88 | 3.45 | 34.84 | 3.71 | 34.18 | 4.94 | 13.00 |

K alkali is more likely to hinder the hydration of the Al phase than Na alkali, which may be related to the formation of potassium gypsum ($K_2Ca(SO_4)_2 \cdot H2O$). In this study, it is found that the slurry with K alkali shows an obvious thickening phenomenon. Some studies [85,86] have shown that potassium gypsum was formed in cement pastes with high K alkali content and had a thickening effect, as shown in Equation (5), but there was no similar compound in those with high Na alkali content. Furthermore, although common compounds formed in sulfate can promote the hydration of $C_3A$, potassium gypsum can reduce the hydration of $C_3A$ [85].

$$K_2SO_4 + CaSO_4 \cdot 2H_2O = K_2Ca(SO_4)_2 \cdot H_2O + H_2O \tag{5}$$

Furthermore, as noted in Table 5, compared with L0, alkali sulfates can promote the transfer of Al atom in hydration products of LHPC pastes to the C-S-H chain to some extent, but the promotion effect decreases with the increase of alkali sulfates content, which is consistent with the findings of relevant research [87]. For instance, when the alkali sulfates content increases from 0.4% to 0.8%, the content of Al (4) in LHPC pastes increases from 31.16% to 32.28%, while when the alkali sulfates content increases to 1.2%, the content of that only increases to 31.48%. In addition, when the alkali sulfates content is 1.2%, the atomic substitution ratios of Al are 34.84% and 31.48%, respectively. This indicates that K alkali is more likely to promote the transfer of Al atom to the C-S-H chain than Na alkali. The same conclusion has been reached in the study of the influence of NaOH on the structure of Al phase in OPC [17]. However, the substitution of an Al atom for a Si atom in the C-S-H chain will cause the recombination of cations and water molecules in the vicinity leading to local charge defects [88,89]. Some studies also have shown that such substitution weakens the mechanical properties of C-S-H [90]. Therefore, the substitution properties of Al atoms in the C-S-H chain may be another reason for alkali sulfates to promote the shrinkage of cement-based materials.

Moreover, Table 5 indicates that alkali sulfates can promote the transfer of Al atom to hydration products of AFt, AFm and TAH with Al (6), in which the promotion effect of K alkali is more profound than that of Na alkali. When the alkali sulfates content is 1.2%, the content of Al (6) in LHPC pastes with K alkali is 52.12%, while that with Na alkali is 49.71%. In Al (6), more than 60% of the Al atom exists in AFt, which is mainly caused by the presence of $SO_4^{2-}$ [91], as shown in Equation (6). In addition, it is worth noting that the content of AFt in LHPC pastes with K alkali is lower than that with Na alkali, which may be due to the formation of potassium gypsum by consuming some $SO_4^{2-}$. It is well-known that AFt can act as an expansive skeleton and limit the shrinkage of cement-based material

to a certain extent. Thus, the lower AFt content may be one of the reasons that K alkali is more likely to promote the shrinkage of cement-based materials than Na alkali.

$$C_6A\bar{S}_3H_{32} + 4ROH \rightarrow C_4A\bar{S}H_{12} + 2R_2SO_4 + 2CH + 20H \tag{6}$$

### 3.6. The Hydration Products of Si Phase of Cement-Based Materials with Different Types of Alkali Sulfates

Generally, according to the difference of bridge oxygen number when silicate oxygen is connected with other Si atoms, the chemical environment of Si is expressed as $Q^0$, $Q^1$, $Q^2$, $Q^3$ and $Q^4$. $Q^0$, $Q^1$ and $Q^2$ mainly exist in pure cement paste, and the corresponding chemical shifts in NMR spectra are −68 ppm~−76 ppm, −76 ppm~−82 ppm and −82 ppm~−88 ppm, respectively. Due to the substitution of Al, $Q^2$ (Al) still exists, its chemical shifts move to the low field, near −82 ppm [92,93]. This section also takes the LHPC system as an example.

The $^{29}$Si MAS NMR curves of cement pastes containing different contents and types of alkali sulfates at 3 d are shown in Figure 16. In the four groups of samples, $Q^0$ shows two obvious peak positions, which represent $C_2S$ and $C_3S$ phases in cement [94], while the peak intensity of the $C_3S$ phase is significantly lower than that of the $C_2S$ phase, which confirms that the hydration activity of the $C_3S$ phase is higher than that of the $C_2S$ phase. The $^{29}$Si MAS NMR curve is made up of a series of signals. The deconvolution of the spectral lines can effectively distinguish the relative intensity I ($Q^n$) of $Q^n$. According to the relative strength, the hydration degree of cement paste ($\alpha$), average chain length (ACL) of C-S-H gel and the four-coordination Al in C-S-H can be calculated.

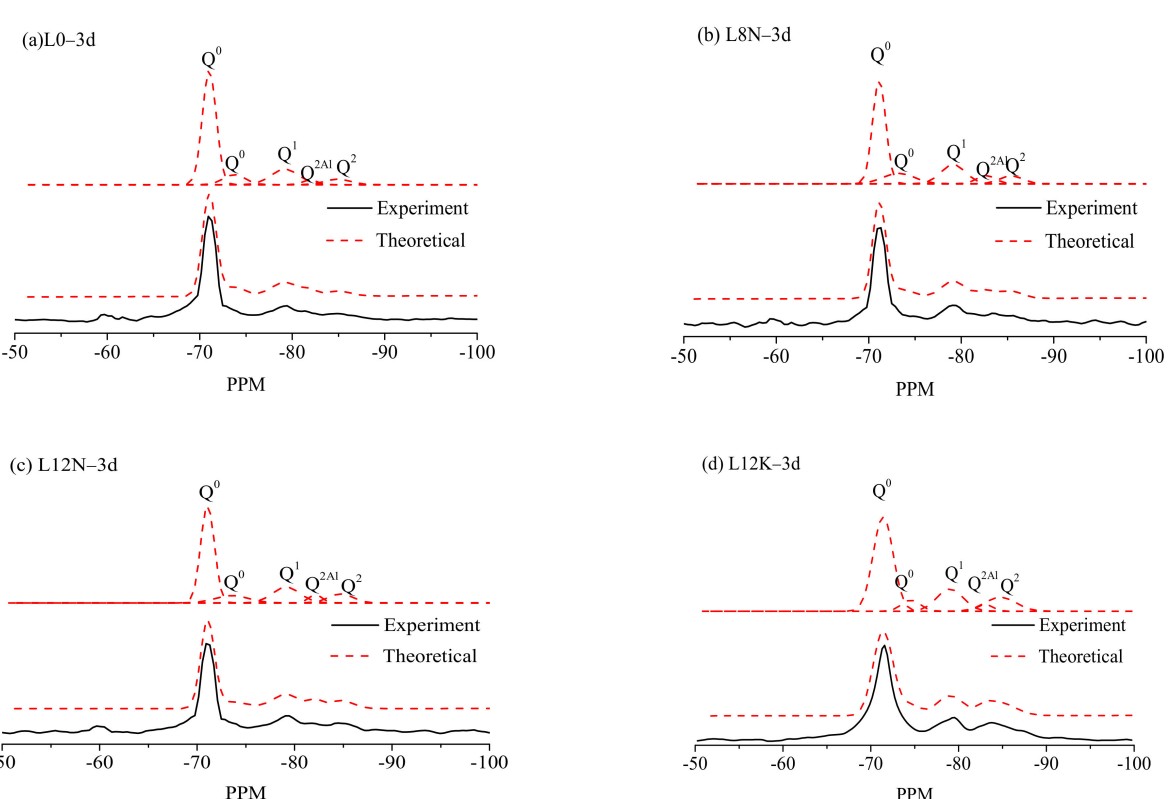

**Figure 16.** $^{29}$Si MAS NMR curves of cement pastes with different contents and different types of alkali sulfates at 3d. (**a**) LHPC pastes with alkali content of 0.4% at 3d; (**b**) LHPC pastes with alkali content of 0.8% (added with $Na_2SO_4$) at 3d; (**c**) LHPC pastes with alkali content of 1.2% (added with $Na_2SO_4$) at 3d; (**d**) LHPC pastes with alkali content of 1.2% (added with $K_2SO_4$) at 3d.

The composition of the Si phase in LHPC pastes with different contents and types of alkali sulfates at 3 d are shown in Table 6. It can be seen that the hydration degree of L0-3d is only 26.35% at 3 d. With the increase in alkali sulfates content, the hydration of LHPC pastes is improved, but it is only kept at about 32%. This indicates that the hydration of LHPC pastes is slow in the early stage and that the improvement effect of alkali sulfates is limited, which may be related to the special mineral composition of LHPC pastes.

**Table 6.** Composition of Si phase in cement pastes with different contents and types of alkali sulfates at 3 d.

| Sample | Composition of Si Phase/% | | | | | ACL | $\alpha$/% | Al [4]/Si |
|---|---|---|---|---|---|---|---|---|
| | $I(Q^0)_{C2S}$ | $I(Q^0)_{C3S}$ | $I(Q^1)$ | $I[Q^2(Al)]$ | $I(Q^2)$ | | | |
| L0-3d | 64.63 | 9.01 | 17.65 | 2.78 | 5.92 | 3.14 | 26.35 | 5.28 |
| L8N-3d | 54.71 | 13.06 | 18.80 | 6.20 | 7.24 | 3.76 | 32.24 | 9.61 |
| L12N-3d | 57.01 | 10.28 | 18.55 | 4.71 | 9.45 | 3.78 | 32.71 | 7.20 |
| L12K-3d | 57.73 | 9.77 | 16.20 | 5.04 | 11.26 | 4.32 | 32.50 | 7.75 |

Furthermore, alkali sulfate promoted the increase in ACL of C-S-H, and the promotion effect of K alkali is higher than that of Na alkali. Compared with ACL of C-S-H in L0-3d, when the alkali content was 1.2%, Na alkali could increase the ACL of LHPC paste by 19.7%, while K alkali could help to increase the ACL by 37.6%. This shows that K alkali can promote the polymerization degree of C-S-H more easily than Na alkali. However, the increase in polymerization degree of C-S-H can cause the shrinkage of the C-S-H chain, which may be one of the reasons for the different shrinkage of cement-based materials caused by different types of alkali. In general, the increase in ACL indicates the decrease in Ca/Si in C-S-H. This is consistent with the study of Bu and Weiss [95], who found that the Ca/Si ratio of C-S-H decreased from 1.64 to 1.42 with the alkali content ranging from 0.21% to 1.01%.

In addition, alkali sulfate can promote the substitution of Al in C-S-H, but when the alkali content exceeds a certain amount, the promotion effect decreases. Compared with L0-3d, ratios of Al [4]/Si in LHPC paste with Na alkali of 0.8% and 1.2% were increased by 82.0% and 36.4%, respectively. However, the promotion effect of K alkali is more profound than that of Na alkali: when the alkali content was 1.2%, the ratio of Al [4]/Si in LHPC paste with Na alkali increased by 36.4%, while that with K alkali increased by 46.8%. This indicates that K alkali may be likely to promote the transfer of Al atom into C-S-H chain.

*3.7. The SEM Spectrum Pictures of Cement Hydration Products with Different Types of Alkali Sulfates*

An SEM map of LHPC and MHPC mortars is given in Figures 17 and 18. Curly leaf-like C-S-H hydration products can obviously be seen, and large CH is embedded in a large number of hydration products in L0-3d. The same morphology can be observed in M0-3d. When the total alkali content increases to 1.2%, Na alkali makes the curly leaf-like C-S-H hydration products obviously refined, and they became invisible in LHPC mortars; Na alkali, in turn, accelerates the transformation of C-S-H morphology from curly leaf-like to needles, for example, in MHPC mortars. In addition, the small CH chips are embedded in the hydration products in LHPC and MHPC mortars. There are no obvious curly leaf-like C-S-H hydration products in LHPC and MHPC mortars with K alkali of 1.2% content, and no CH is observed in this field of view.

Figure 19 shows the morphology of hydration products in OPC mortars. Unlike LHPC and MHPC mortars, the long fibrous-like C-S-H and the large lamellar CH are embedded in the hydration products of P0-3d, which is consistent with the morphology observed in Reference [8]. When the total alkali content increases to 1.2%, Na alkali disintegrates the fibrous like C-S-H, turning into amorphous C-S-H gel, no obvious CH are observed. In OPC mortars with K alkali of 1.2% content, C-S-H are flakes and stacks with each other, and no plate CH are observed.

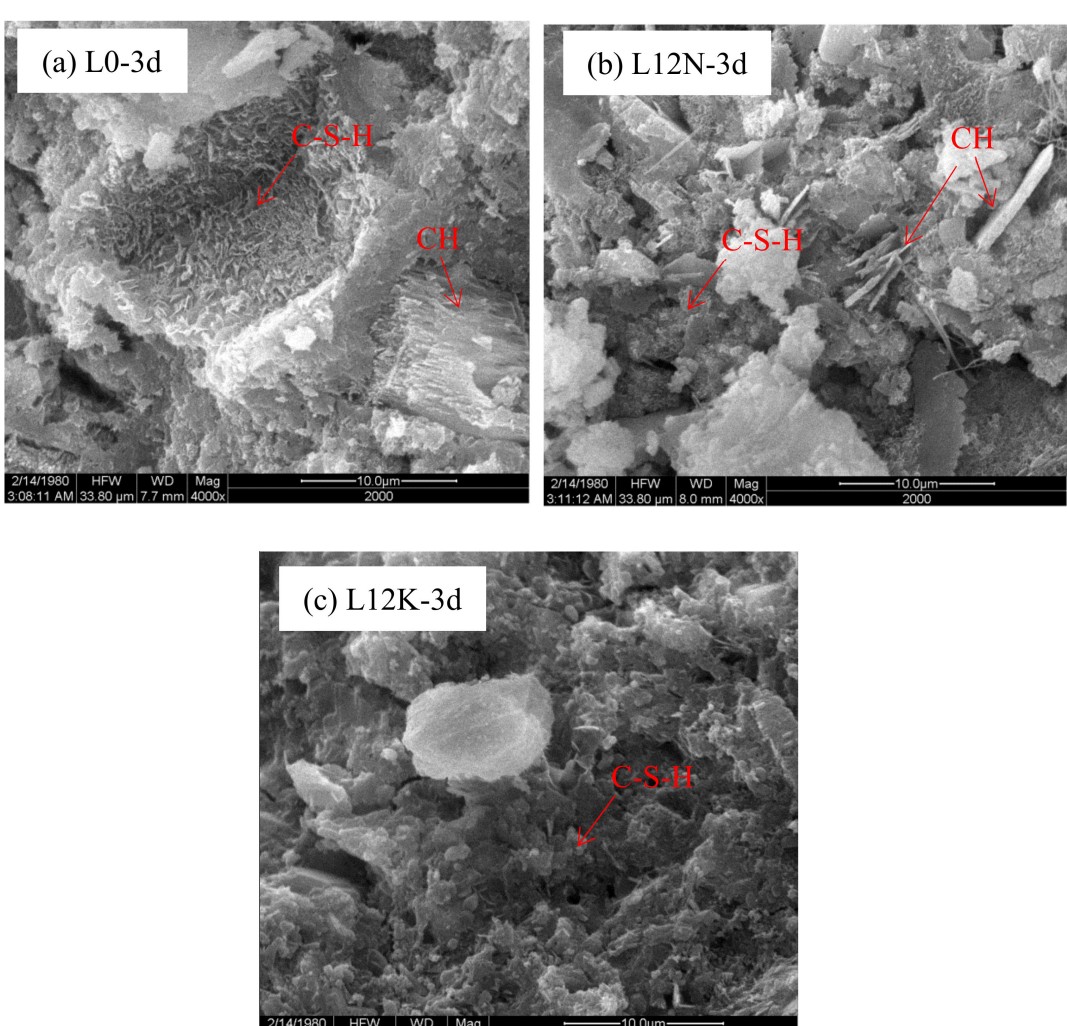

**Figure 17.** SEM map of LHPC mortars with different content and different types of alkali sulfates (4000×). (**a**) LHPC pastes with alkali content of 0.4% at 3d; (**b**) LHPC pastes with alkali content of 1.2% (added with $Na_2SO_4$) at 3d; (**c**) LHPC pastes with alkali content of 1.2% (added with $K_2SO_4$) at 3d.

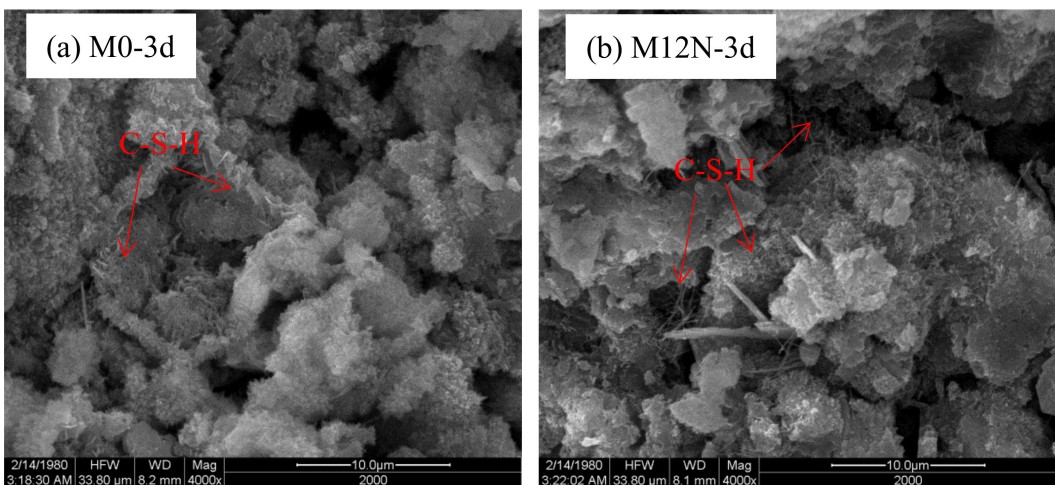

**Figure 18.** *Cont.*

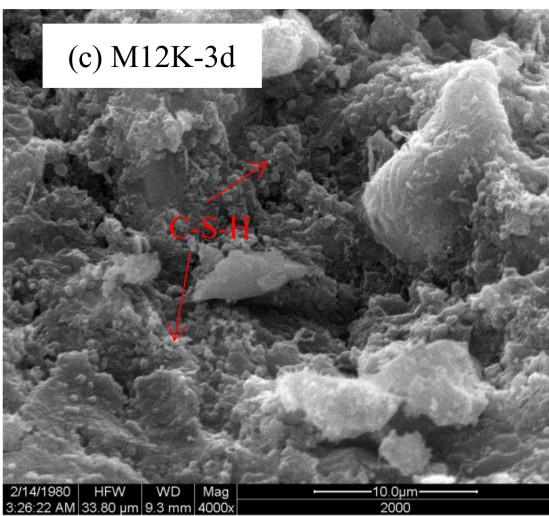

**Figure 18.** SEM map of MHPC mortars with different content and different types of alkali sulfates (4000×). (**a**) MHPC pastes with alkali content of 0.4% at 3d; (**b**) MHPC pastes with alkali content of 1.2% (added with $Na_2SO_4$) at 3d; (**c**) MHPC pastes with alkali content of 1.2% (added with $K_2SO_4$) at 3d.

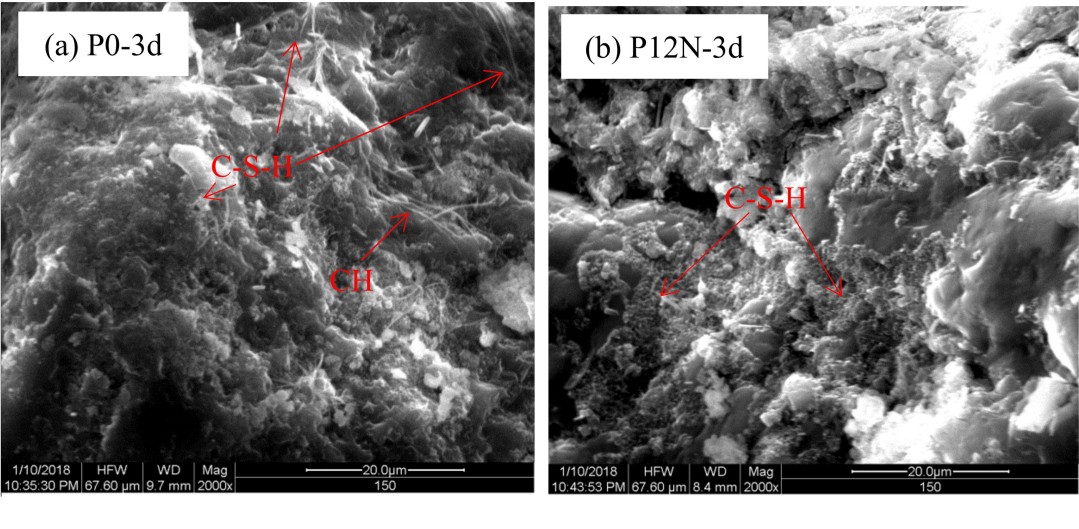

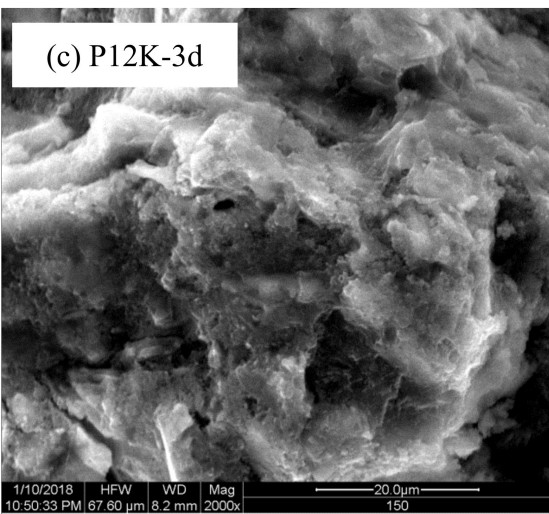

**Figure 19.** SEM map of OPC mortars with different contents and different types of alkali sulfates (2000×). (**a**) OPC pastes with alkali content of 0.4% at 3d; (**b**) OPC pastes with alkali content of 1.2% (added with $Na_2SO_4$) at 3d; (**c**) OPC pastes with alkali content of 1.2% (added with $K_2SO_4$) at 3d.

As mentioned above, it can be concluded that the morphology of hydration products of LHPC, MHPC and OPC mortars are quite different due to the difference of mineral compositions. Alkali sulfates can refine the hydration products C-S-H and CH, which is consistent with the findings by Burrows et al. [8]. However, it had been pointed out that alkali sulfates could promote the formation of lathy-like C-S-H in cement-based materials [86,96], which are not observed in this study. Furthermore, the typical morphology of C-S-H and CH are almost invisible in mortars with K alkali, which may be due to the profound promotion effect of K alkali, making large number of hydration products in cement-based materials stack with each other.

## 4. Conclusions

(1) Alkali sulfates could promote the autogenous shrinkage and dry shrinkage of LHPC mortars, MHPC mortars and OPC mortars to a large extent. The promotion effect of alkali sulfates on self-autogenous was greater than that on dry shrinkage, and the promotion effect of K alkali was greater than that of Na alkali. Based on the ratio of autogenous shrinkage and dry shrinkage, the drying shrinkage of LHPC caused by alkali sulfates could be greatly reduced by early curing.

(2) Alkali sulfates mainly promoted the hydration degree of cement-based materials by shortening the induction period and increasing the maximum rate of hydration, in which the promotion effect of K alkali was greater than that of Na alkali, but the promotion effect weakened with the increment of alkali sulfates content. The promotion effect of alkali sulfates on the hydration of LHPC was greater than that of MHPC and OPC, which may be related to different $C_3A$ content.

(3) Based on the results of pore structure and Ds of cement-based material, the mechanism of alkali sulfates promoting the shrinkage of cement-based materials was that alkali sulfates could refine the pore structure and increase the number of pores whose diameters are smaller than 50 nm that had a great influence on the shrinkage. K alkali more profoundly promoted the shrinkage of cement pastes than Na alkali. Ds could better characterize the shrinkage characteristics of cement-based materials than traditional pore structure parameters.

(4) The results of NMR showed that K alkali can promote the transfer of Al atoms into the C-S-H chain to a great extent, which may weaken the mechanical properties of C-S-H and reduce the content of AFt as an expansive skeleton.

(5) The morphology of C-S-H in LHPC and MHPC mortars was curly leafed, but it was fibrous in OPC mortars. Alkali sulfates could refine the structure of C-S-H and CH. Compared with Na alkali, the refining effect of K alkali was obvious due to the profound promotion of K alkali on the hydration degree of cement-based materials.

**Author Contributions:** Y.L. and H.Z. mostly contributed to the design of the manuscript. M.H. and H.Y. were carried out data collection and processing. K.J. and K.X. were involved in the statistical analysis. S.T. revised the paper. All authors have read and agreed to the published version of the manuscript.

**Funding:** Authors are thankful to Basal Research Fund of Central Public Welfare Scientific Institution of China under grant of CKSF2017034/GC, CKSF2017032/GC and CKSF2019166/JC, the Opening Funds of State Key Laboratory of Building Safety and Built Environment and National Engineering Research Center of Building Technology under grant of BSBE2020–1, Jiangsu Province Natural Science Foundation under grant of BK20181187, Opening Project of State Key Laboratory of Green Building Materials under grant of 2019GBM05.

**Institutional Review Board Statement:** Not applicable.

**Informed Consent Statement:** Not applicable.

**Data Availability Statement:** Not applicable.

**Acknowledgments:** The authors would like to thank all three anonymous referees for their constructive comments and suggestions.

**Conflicts of Interest:** The authors declare no conflict of interest.

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
