# Peer review of "Influence of Different Alkali Sulfates on the Shrinkage, Hydration, Pore Structure, Fractal Dimension and Microstructure of Low-Heat Portland Cement, Medium-Heat Portland Cement and Ordinary Portland Cement"

_fractalfract, doi:10.3390/fractalfract5030079_

Round 1

Reviewer 1 Report

Please see the file attached.

Reviewer 2 Report

An interesting read although it will require some polishing to ensure the English is both accurate and readily understandable. 

The first sentence of the introduction requires rethinking and would be better aimed at the importance of durability in infrastructure generally rather than any specific national requirement. 

Once such checks and changes are made I would be happy to recommend publication.

Round 2

Reviewer 1 Report

All the comments and suggestions have been adressed. The paper has been improved and can be published.